# Nocturnal Water Use Partitioning and Its Environmental and Stomatal Control Mechanism in *Caragana korshinskii* Kom in a Semi-Arid Region of Northern China

Wei Li [1,2], Yu Zhang [3,4,5,*], Nan Wang [1], Chen Liang [1,2], Baoni Xie [1,2], Zhanfei Qin [1,2], Ying Yuan [6] and Jiansheng Cao [7,*]

1   School of Land Science and Space Planning, Hebei GEO University, Shijiazhuang 050031, China; weil87land@hgu.edu.cn (W.L.); wn305099@163.com (N.W.); liangchen214@mails.ucas.ac.cn (C.L.); xbn-feya@nwafu.edu.cn (B.X.); zhanfeiqin@163.com (Z.Q.)
2   International Science and Technology Cooperation Base of Hebei Province: Hebei International Joint Research Center for Remote Sensing of Agricultural Drought Monitoring, Hebei GEO University, Shijiazhuang 050031, China
3   School of Geographical Sciences, Hebei Normal University, Shijiazhuang 050024, China
4   Hebei Key Laboratory of Environmental Change and Ecological Construction, Shijiazhuang 050024, China
5   Hebei Technology Innovation Center for Remote Sensing Identification of Environmental Change, Shijiazhuang 050024, China
6   School of Urban Geology and Engineering, Hebei GEO University, Shijiazhuang 050031, China; yuanyingson@hgu.edu.cn
7   Center for Agricultural Resources Research, Institute of Genetics and Developmental Biology, Chinese Academy of Sciences, Shijiazhuang 050022, China
*   Correspondence: yuzhang89@bnu.edu.cn (Y.Z.); caojs@sjziam.ac.cn (J.C.)

**Abstract:** As an important aspect of plant water consumption, nocturnal water use ($E_n$) behavior provides reliable information on the effect of plantation carbon and water budgets at stand and regional scales. Therefore, quantifying $E_n$ and its environmental and stomatal controlling mechanisms is urgent to establish adaptation strategies for plantation management in semiarid regions. With the help of the sap flow technique, our study investigated the seasonal variations in canopy transpiration and canopy conductance in a *Caragana korshinskii* Kom plantation. Environmental variables were measured concurrently during the growing seasons of 2020 and 2021. The results indicated that the average $E_n$ values were 0.10 mm d$^{-1}$ and 0.09 mm d$^{-1}$, which accounted for 14% and 13% of daily water use, respectively, over two years. The proportions of nocturnal transpiration ($T_n$) to $E_n$ were approximately 49.76% and 54.44%, while stem refilling ($R_e$) accounted for 50.24% and 45.56% of $E_n$ in 2020 and 2021, respectively, indicating that *C. korshinskii* was able to draw on stored stem water to support transpiration. $E_n$ was predominantly affected by nocturnal canopy conductance ($G_{c-n}$), air temperature ($T_{a-n}$) and wind speed ($u_{2-n}$). In contrast, $G_{c-n}$ and $T_{a-n}$ explained the highest variation in $T_n$ and nocturnal vapor pressure ($VPD_n$), and $u_{2-n}$ explained the highest variation in $R_e$. Total effects of the five environmental and stomatal variables explained 50%, 36% and 32% of $E_n$, $T_n$ and $R_e$ variation, respectively. These findings could enable a better understanding of nocturnal water use dynamics and their allocation patterns in *C. korshinskii* plantations on the Bashang Plateau. Moreover, our results reveal the water use strategies of artificial shrubs and highlight the importance of incorporating nocturnal water use processes into large-scale ecohydrological models in semiarid regions.

**Keywords:** nocturnal water use; nocturnal transpiration; stem refilling; canopy stomatal conductance; *Caragana korshinskii* Kom

## 1. Introduction

Nocturnal water use is defined as the sap flux dynamics that occur at night with incomplete stomatal closure, and is different to diurnal water use in a single component.

During the past two decades, nocturnal water use has been reported across a range of forest species and habitats; and the average contribution of nocturnal water use to daily water use is approximately 12% in diverse woody species, and reaches up to 30–60% in water-limited environments [1–5]. The occurrence, pattern and alteration of nocturnal water use play important roles in adjusting predawn disequilibrium between leaves and soil [6], reducing hydraulic redistribution within dry rhizosphere soils [7], avoiding the delay between carbon assimilation and stomatal opening in the early morning hours [8,9] and facilitating oxygen and nutrient uptake for tree growth [10,11]. Therefore, nocturnal water use is regarded as a special water use strategy that enables plants to adapt to different environments, and the accurate quantification of nocturnal water use and its contribution to total daily water use is important for understanding the carbon and water budgets of forest ecosystems [12]. However, there is limited research on the nocturnal water use behavior of planted shrubs in arid and semiarid regions, which limits our understanding of the water use strategy of shrubs, and of future changes in ecosystem function in these regions.

Nocturnal water use can be characterized as nocturnal transpiration from leaves and the water required for stem refilling, separately or synergistically [13–15]. Nocturnal transpiration is driven by leaf-to-air vapor pressure differences under the influence of nocturnal stomatal openness. It might help to promote nutrient absorption via the mass flow of water and carbon fixation in the early morning [16]. Stem refilling is an effective type of internal water storage, that functions to replenish the large amount of water consumed during diurnal transpiration. Water absorption by the roots during the night is needed to overcome xylem and stem water deficits that occur in the daytime. Precisely identifying nocturnal transpiration and stem refilling components from continuous nocturnal water use will contribute to developing a better understanding of physiological responses to external environmental changes. Several methods have been proposed to distinguish nocturnal transpiration and stem refilling from nocturnal water use, such as the Penman–Monteith model [17], a flow model based on a resistance network [18], a time series model with one or more exogenous inputs [19,20] and a forecasted refilling method regarding determination-based vapor pressure deficits (*VPD*) [13,21]. The forecasted refilling method, conducted by analyzing the slope of a daily sap flux curve, has been proven to be a simple and useful fraction method, and has been successfully applied on rainless nights in natural [1,2], planted [22,23] and urban forests [5].

The reported crucial environmental drivers of nocturnal water use include nocturnal VPD, temperature, wind speed and soil water content [7,22,24]. High nocturnal water use usually occurs when air temperatures and soil water content are high. Additionally, nocturnal water use in response to environmental factors is achieved through adjustments to nocturnal stomatal conductance. High nocturnal water use is regularly associated with high nocturnal stomatal conductance, which tracks VPD-induced variations in stomatal apertures [25]. The environmental and stomatal control mechanisms of nocturnal transpiration and stem refilling are generally different, according to prior studies. The concurrent occurrence of stomatal openness and vapor pressure differences between the atmosphere and leaf surface provide the necessary conditions for nocturnal transpiration [5]. Therefore, nocturnal transpiration is usually positively correlated with VPD and wind speed, while stem refilling is commonly correlated to biological and physiological variations, such as plant size, anatomical features, plant water status, etc. [26]. However, similar stem refilling proportion results were reported for plants of the same species but different individual sizes in planted forests in Mexico [27]. Therefore, the environmental and stomatal mechanism of nocturnal water use is still uncertain, and a more advanced understanding of the total effect of concurrent environmental and stomatal controls on nocturnal water use and its fractions is needed.

As a typical shrub type in arid and semiarid regions, *Caragana korshinskii* Kom has commonly been used since the implementation of the Beijing–Tianjin Sand Source Restoration Project in 2000 for establishing protective shrub plantations on the Bashang Plateau. Due to their strong drought tolerance, *C. korshinskii* plantations play an important role in

windbreak and sand fixation, soil and water conservation, soil restoration and ecological protection. Major research has been conducted on *C. korshinskii* plantations to understand their daily water use and their capacity to adapt to water-limited environmental conditions [28–30]. However, the ecological importance of nocturnal water use dynamics in *C. korshinskii* plantations is poorly understood, and under soil water scarcity the responses of nocturnal water use, nocturnal transpiration and stem refilling to concurrent environmental and stomatal variables remain fully unknown. Therefore, we hypothesize that (1) nocturnal water use occurs in *C. korshinskii* plantations and has a great impact on daily water loss; (2) the environmental and stomatal controls of nocturnal water use and its fractions (nocturnal transpiration and stem refilling) may be subject to different influences. The objectives of this study are as follows: (1) to analyze the nocturnal water use dynamic and its contribution to the daily water use of *C. korshinskii*, (2) to determine the allocation of the nocturnal transpiration and stem refilling of *C. korshinskii* and (3) to explore the total effects of concurrent environmental and stomatal controls on nocturnal water use, nocturnal transpiration and stem refilling.

## 2. Materials and Methods

### 2.1. Site Description

The research area is situated at the northern Bashang Plateau, Kangbao County, which belongs to the northwest of the Beijing–Tianjin–Hebei Region of China (Figure 1). The climate of the region is arid and cold with mean annual precipitation of 330.0 mm and mean annual temperature of 2.3 °C (1980–2020). However, the potential evapotranspiration is over 850 mm. The mean annual wind speed is 3.15 m s$^{-1}$. The field site located at the Kangbao pasture region (114°48′ E, 42°07′ N, altitude 1305 m) is composed of planted forests of pure *C. korshinskii*, which were planted according to the Beijing–Tianjin Sandstorm Source Sontrol Project. The *C. korshinskii* plantation is about 20 years old, with an average height of 179.28 ± 30.74 cm and stem basal diameter at stem base (10–15 cm above the ground) of 2.38 cm. The soil is mainly sandy loam with a high sand content of 52.89% and low clay content of 7.86% [31].

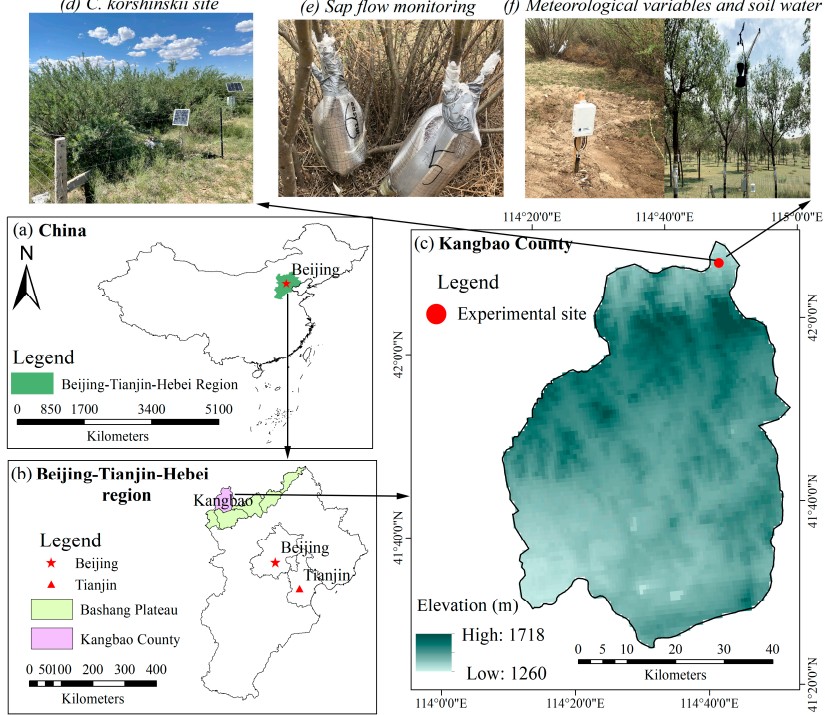

**Figure 1.** Geographical research sites in this study, including (**a–c**) Beijing–Tianjin–Hebei region and the Bashang Plateau of Hebei province, (**d**) *Caragana korshinskii* Kom site in Kangbao County, (**e,f**) sap flow measurement and meteorological variables, soil water monitoring.

### 2.2. Materials and Experimental Design

2.2.1. Materials

We selected a sample site for the study and established a 20 m × 20 m fenced enclosure for purposes of monitoring. Based on the distribution of stem base diameter (SBD), nine sample stems (SDB = 2.68, 2.96, 3.38, 3.62, 3.66, 3.76, 4.14, 4.24 and 4.38 cm) of healthy and non-stressed *C. korshinskii* were selected for the 2020 and 2021 experimental periods. We assumed that the whole cross-section of these stems in *C. korshinskii* were conductive, sapwood area ($A_s$) which was then estimated as a stem basal cross-sectional area. The total sapwood area in the experimental plot was 6877.70 cm$^2$.

2.2.2. Environmental Variables

Meteorological parameters were measuring at a height of 2 m in a nearby clear space with an Onset HOBO U21 automatic weather station (Onset Computer Corp., Bourne, MA, USA) and included the following: photosynthetically active radiation ($PAR$, μmol m$^{-2}$ s$^{-1}$), air temperature ($T_a$, °C), relative humidity ($RH$, %), wind speed ($u_2$, m s$^{-1}$) and precipitation ($P$, mm). Vapor pressure deficits ($VPD$, kPa) were calculated from $T_a$ and $RH$ according to Campbell and Norman [32]. These meteorological data were available as 10 min averages from 30 s samples and stored using CR1000 data loggers (Campbell Scientific, Logan, UT, USA).

The soil water content ($SWC$, m$^{-3}$ m$^{-3}$) was measured at five depths (i.e., 5, 15, 30, 50, 80 cm) with an EC-5TE sensor (Decagon, Inc. Pullman, WA, USA) and recorded at 30 min internals by CR1000 data loggers (Campbell Scientific, Logan, UT, USA).

Relative extractable soil water content ($REW$) was calculated using averaged $SWC$ across 0–100 cm:

$$REW = \frac{SWC - SWC_{min}}{SWC_{max} - SWC_{min}}$$

where $SWC_{min}$ and $SWC_{max}$ are the minimum and maximum daily average soil water content, respectively, during the two whole years.

2.2.3. Sap Flow Measurements

We measured xylem sap flux on stems using thermal dissipation probes (TDPs) during two growing seasons in 2020 and 2021 (1 May to 30 September). Nine pairs of Granier-type probes (TDP10, Dynamax Inc., Houston, TX, USA) consisting of a copper-constantan thermojunction were inserted at a depth of 10 mm into the xylem sapwood of stems of nine shrubs at a height of 40 cm above the ground on the northern side. In order to protect the probes to reduce solar heating and extraneous thermal gradients, an aluminum foil shield wrap was applied for each pair of Granier-type probes. Thermal insulation cotton and aluminum foil were placed on the ground around the probe to reduce the influence of ground temperature on its measurements. Additionally, for the homogenous and dense plantation stands, the influences of natural thermal gradients on sap flow measurements could be assumed to be negligible [22,33]. The detail procedure for measuring sap flow velocity can be found in [31]. The sap flow velocity ($SF$, mL cm$^{-2}$ min$^{-1}$) could be calculated from the measured temperature difference at 30 min intervals (CR1000, Campbell Scientific Inc., Logan, UT, USA) according to Granier [34]:

$$SF = 0.0119 \times \left( (\Delta T_{max} - \Delta T)/\Delta T \right)^{1.231} * 60$$

$$E = \overline{SF} \times \left( A_{si}/A_g \right) \times 600$$

where $\Delta T$ (°C) is the temperature difference between the two probes at any given time, and $\Delta T_{max}$ (°C) is the maximum temperature difference between sensors, which was determined as the maximum value of daily $\Delta T_{max}$ over a 9-day period to avoid underestimation of the nighttime sap flow. (1) A linear regression of local maximal $\Delta T_{max}$ determined by a 9-day moving window and time (day) was performed and local $\Delta T_{max}$ values below the linear regression line were eliminated; (2) A new linear regression was then made based

on the remaining $\Delta T_{max}$ points; (3) finally the "real $\Delta T_{max}$" was recalculated by the new linear regressions [35,36]. As there was only one probe per sample stem, the azimuthal and radial variation in sap flow velocity were not taken into account in this study. We assume that these variations were low [23,37]. $E$ is canopy transpiration per area of ground (mm h$^{-1}$); $\overline{SF}$ is the average sap flow velocity of sampled shrubs (ml cm$^{-2}$ min$^{-1}$), $A_g$ is the ground surface area of the studied plots (400 m$^2$), and $A_{si}$ is the total sapwood area in the studied plot.

Daily $SF$ and $E$ were further divided into diurnal $SF/E$ ($SF_d/E_d$) and nocturnal $SF/E$ ($SF_n/E_n$) based on the value of $PAR$. Diurnal $SF_d/E_d$ was defined as a $PAR$ value greater than 5 μmol m$^{-2}$ s$^{-1}$, while nocturnal $SF_n/E_n$ was correspondingly defined as a $PAR$ value less than 5 μmol m$^{-2}$ s$^{-1}$ or equal to 0. Therefore, nocturnal $SF_n/E_n$ ranged from 19:30 to 4:30 in May, 20:00 to 4:30 in June, 20:00 to 4:30 in July, 19:00 to 5:00 in August, 18:30 to 5:30 in September during the growing season in 2020 (Figure 2a). In 2021, nocturnal $SF_n/E_n$ ranged from 19:30 to 4:30 in May, 20:00 to 4:00 in June, 19:00 to 4:30 in July, 19:30 to 5:00 in August and 18:30 to 5:30 in September during the growing season (Figure 2b). To differentiate between the contributions of nocturnal transpiration ($T_n$) and stem refilling ($R_e$) to $SF_n$, the forecasted refilling method was used [2,21]. Fisher et al. [21] suggested that if no nocturnal water loss occurred, $SF_n$ would gradually continue to fall to zero flow via an exponential decay function. However, we found that $SF_n$ was greater than zero flow during the whole growing season in 2020 and 2021, and decreased at the beginning of the night but then increased until sunrise (Figure 2). It was indicated that the early sloped phase of $SF_n$ mostly consisted of $R_e$ and a nonzero linear phase representing $T_n$ later [2,38]. Therefore, an exponential relationship between $SF_n$ and $VPD_n$ was conducted to fit the first 3 to 5 h:

$$SF_n = a \times \exp(b \times VPD_n)$$

where $a$ and $b$ are fitting parameters. The relationship between $SF_n$ and $VPD_n$ was determined, with $R^2$ consistently > 0.96 (Figure A1) during the growing season in 2020. This approach was only applied for every clear night with low nocturnal $VPD$, so that: (1) the area below the forecasting curve could be considered as sap flow caused by $R_e$, due to lack of strong atmospheric demand for nocturnal water loss; (2) the area above the curve could be considered as sap flow caused by $T_n$.

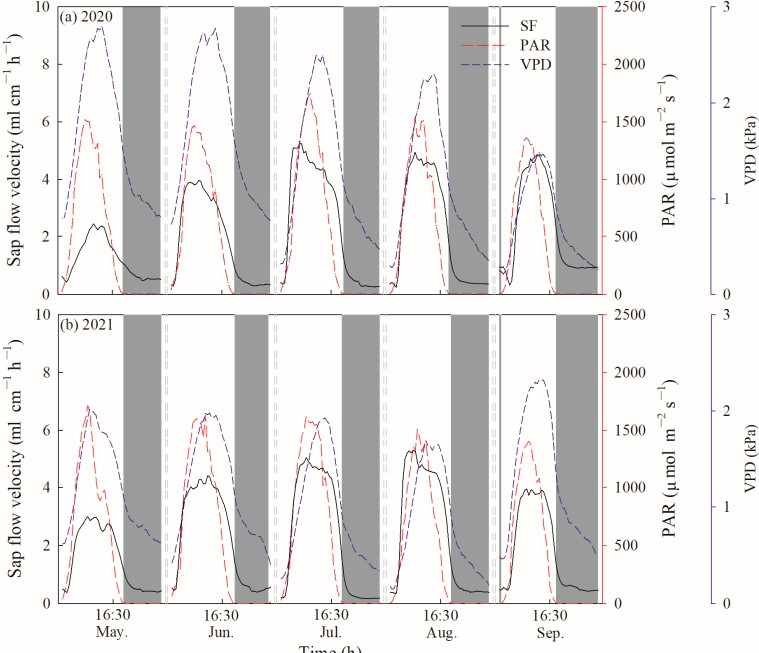

**Figure 2.** Hourly variations in sap flow velocity of *C. korshinskii*, photosynthetically active radiation and vapor pressure deficits.

2.2.4. Canopy Conductance

Since the distribution of the *C. korshinskii* plantation was relative uniform and the canopy was not closed (the largest value of leaf area indices was 0.43 m$^2$ m$^{-2}$), the canopy was well-coupled to the atmosphere [31], The canopy conductance can be calculated by the simplified Penman–Monteith equation [39] using the following formula:

$$G_c = \gamma \lambda E / \rho C_p VPD$$

where $G_c$ is canopy conductance (mm s$^{-1}$), $\gamma$ is the psychometric constant (kPa, °C$^{-1}$), $\lambda$ is latent heat of water vaporization (MJ kg$^{-1}$), $\rho$ is the air density (kg m$^{-3}$) and $C_p$ is the specific heat of the air (MJ kg$^{-1}$ °C$^{-1}$).

Nocturnal $G_c$ ($G_{c-n}$) were also defined as stomatal opening occurring when a *PAR* value was less than 5 μmol m$^{-2}$ s$^{-1}$.

*2.3. Data Analysis*

The differences in the environmental and stomatal variables during the two growing seasons of 2020 and 2021 were tested using a paired-samples *t*-test. Significant differences in $E_n$, $T_n$ and $R_e$ among different months during the growing season were tested using one-way ANOVA at a significance level of $\alpha = 0.05$. Pearson's correlation coefficient and a partial correlation coefficient were used to estimate associated pairwise relationships between $E_n / T_n / R_e$ and all environmental and stomatal variables. Then, linear, polynomial and nonlinear (i.e., exponential growth function, exponential threshold function) regression analyses were conducted to investigate the relationships between $E_n / T_n / R_e$ and major environmental and stomatal variables (i.e., $VPD_n$, $u_{2-n}$, $T_{a-n}$ and $G_{c-n}$). However, these single-variable relationships above might be weak and showed a high degree of scatter, which was due to a strong effect of certain other factors. Therefore, an upper boundary line method was applied to determine these single-variable relationships of $E_n / T_n / R_e$ and $VPD_n$, $u_{2-n}$, $T_{a-n}$ and $G_{c-n}$, without interference from other factors. The study divided $VPD_n$, $u_{2-n}$, $T_{a-n}$ and $G_{c-n}$ into several segments, with intervals of segments of $VPD_n$, $u_{2-n}$, $T_{a-n}$ and $G_{c-n}$ of 0.5 kPa, 1 m s$^{-1}$, 5 °C and 0.5 mm s$^{-1}$, respectively. Then, an upper boundary line was conducted as suitable linear and nonlinear functional forms based on the data of $E_n / T_n / R_e$ of at least one standard deviation greater than the mean $E_n / T_n / R_e$ at each $VPD_n$, $u_{2-n}$, $T_{a-n}$ and $G_{c-n}$ interval. All statistical analyses were conducted using SPSS 21.0 (SPSS Inc., Chicago, IL, USA) and all figures were created using Sigmaplot 11.0 software (Hearne Scientific Software Plc, Melbourne, Australia).

A path coefficient model was developed to quantify the direct and indirect effects of environmental and stomatal variables on $E_n$, $T_n$ and $R_e$, respectively. The initial model included all potential paths according to current knowledge and the above linear/nonlinear analyses. (1) $VPD_n$, $u_{2-n}$, $T_{a-n}$, *REW* and $G_{c-n}$ were considered to directly affect $E_n$, $T_n$ and $R_e$, respectively. Daily *REW* was used to respond to $E_n / T_n / R_e$ since there was no significant difference between diurnal and nocturnal *REW* in this study. (2) $u_{2-n}$ may affect $E_n / T_n / R_e$ by altering $VPD_n$, which may also affect $E_n / T_n / R_e$ by altering $G_{c-n}$. *REW* may affect $E_n / T_n / R_e$ by adjusting nocturnal canopy conductance. The standardized path coefficients were calculated through the maximum likelihood method. Path analysis was conducted using AMOS 22.0 (SPSS Inc., Chicago, IL, USA).

**3. Results**

*3.1. Environmental Conditions*

The average diurnal photosynthetically active radiation (*PAR*) values were 681.16 $\pm$ 201.18 μmol m$^{-2}$ s$^{-1}$ and 744.18 $\pm$ 200.69 μmol m$^{-2}$ s$^{-1}$ during the growing season in 2020 and 2021, respectively. The averaged nocturnal weed speeds ($u_{2-n}$) were 0.76 $\pm$ 0.67 m s$^{-1}$ and 0.80 $\pm$ 0.75 m s$^{-1}$ during the night in 2020 and 2021, respectively (Figure 3a). Nocturnal air temperature ($T_{a-n}$) displayed marked seasonal variations, with ranges of 2.04 to 22.40 °C and −1.89 to 22.61 °C, and mean values of 13.79 $\pm$ 4.38 °C and 13.93 $\pm$ 4.56 °C in

2020 and 2021, respectively. The average nocturnal vapor pressure deficits ($VPD_n$) were $0.60 \pm 0.34$ kPa and $0.62 \pm 0.31$ kPa during the growing season in 2020 and 2021, respectively (Figure 3b). A significant difference was observed in *REW* between 2020 and 2021 (F = 22.299, *p* < 0.000). The daily *REW* ranged from 0.31 to 0.79 and 0.31 to 0.98, with mean values of $0.42 \pm 0.11$ and $0.46 \pm 0.16$, in 2020 and 2021, respectively (Figure 3c).

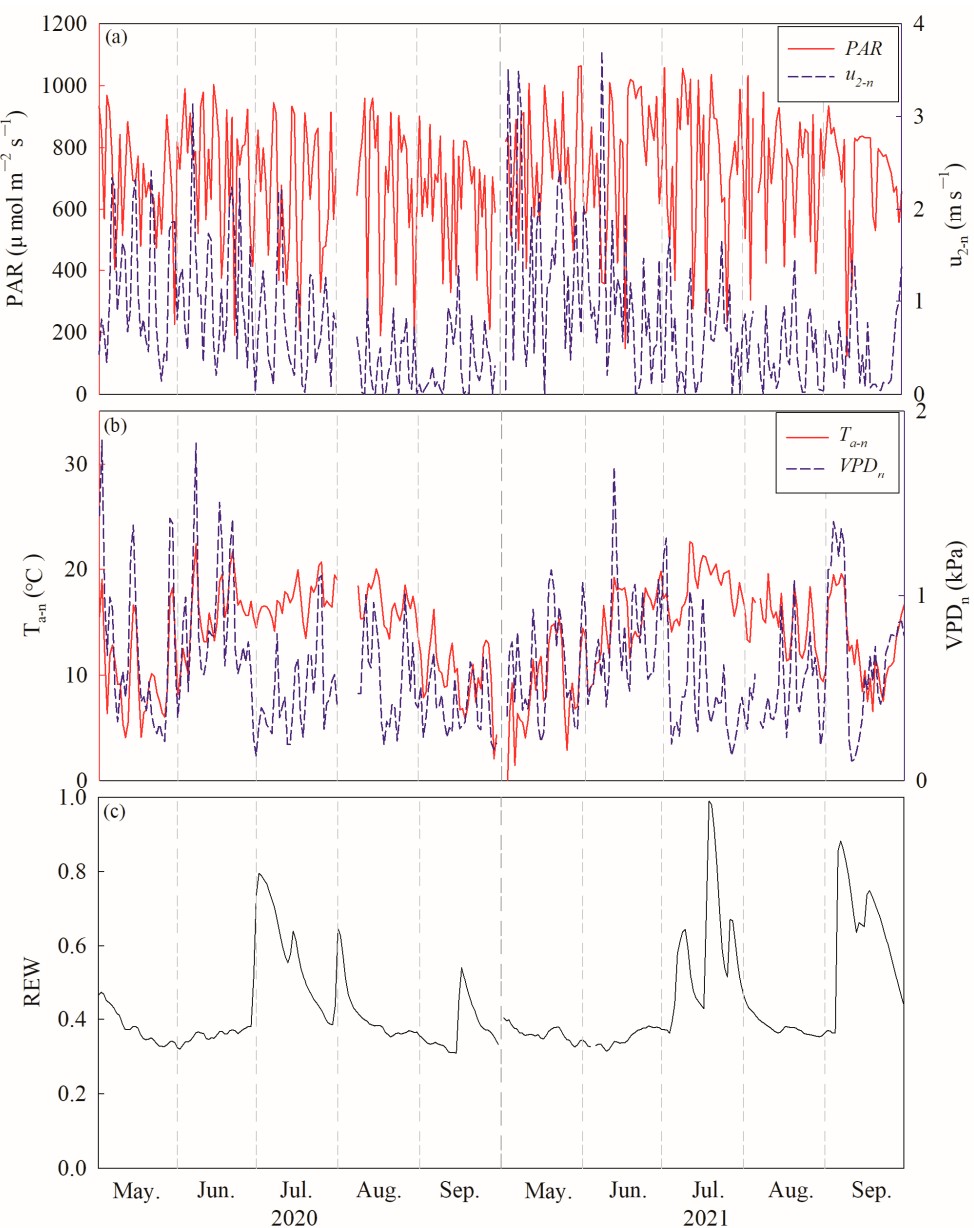

**Figure 3.** Diurnal variations of (**a**) photosynthetically active radiation (*PAR*) and nocturnal wind speed at the height of 2 m ($u_{2-n}$), (**b**) nocturnal air temperature ($T_{a-n}$) and nocturnal vapor pressure deficits and ($VPD_n$) (**c**) soil water content at a depth 0–100 cm (*REW*) in 2020 and 2021.

### 3.2. Variations in Nocturnal Water Use

The annual average $E_d$ values for *C. korshinskii* were $0.64 \pm 0.24$ mm d$^{-1}$ and $0.69 \pm 0.23$ mm d$^{-1}$; meanwhile, the accumulated $E_d$ values over the growing season were 92.51 mm and 103.26 mm in 2020 and 2021, respectively. While the $E_n$ values ranged from 0.01 to 0.61 mm d$^{-1}$ in 2020, with a mean value of $0.10 \pm 0.08$ mm d$^{-1}$, whereas in 2021, $E_n$ ranged from 0.01 to 0.59 mm d$^{-1}$ with a mean value of $0.09 \pm 0.07$ mm d$^{-1}$. The accumulated $E_n$ values over the growing season were 14.36 mm and 14.51 mm in 2020 and 2021, respectively, accounting for 15.48% and 14.04% of the accumulated diurnal water

use and 13.40% and 12.31% of the accumulated daily water use over the same period. The annual average $E_n$:$E_d$ values were 0.18 ± 0.15 and 0.17 ± 0.17, whereas the $E_n$:$E$ values were 0.14 ± 0.09 and 0.13 ± 0.10 in 2020 and 2021, respectively (Figure 4a,b). No significant difference was observed for either $E_n$:$E_d$ or $E_n$:$E$ between 2020 and 2021; however, in both years, $E_n$:$E_d$ and $E_n$:$E$ were significantly higher in May and September than from June to August ($p < 0.000$, ANOVA).

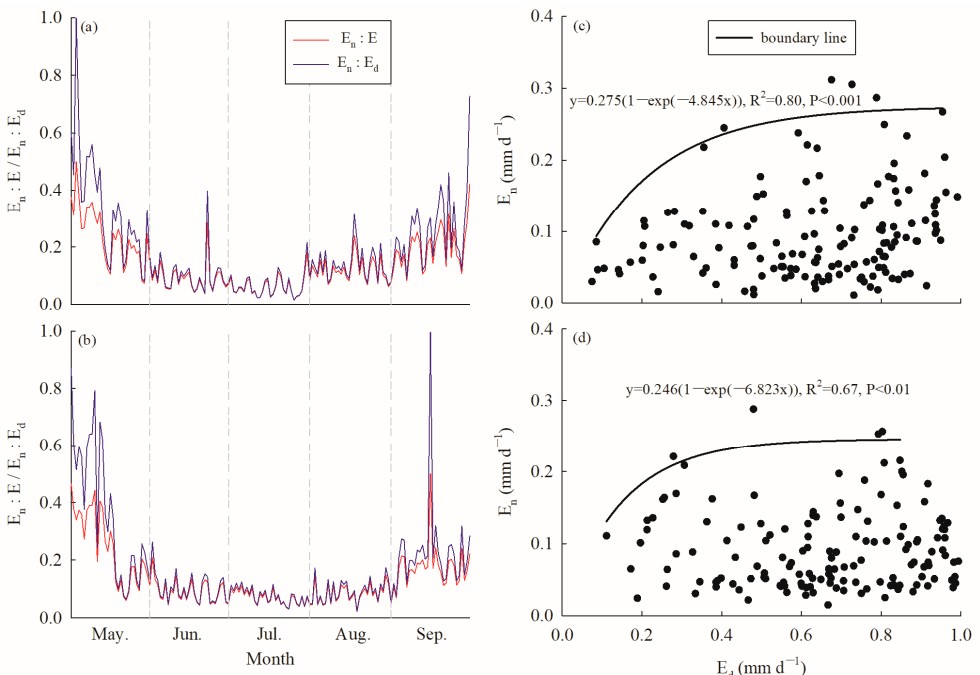

**Figure 4.** Variation in the ratio of nocturnal to diurnal ($E_n$:$E_d$, red line) and nocturnal to daily water use ($E_n$:$E$, blue line) in (**a**) 2020 and (**b**) 2021; the relationship between $E_n$ and $E_d$ (black solid circle) and boundary line analysis (black line) in (**c**) 2020 and (**d**) 2021.

The saturated exponential analysis showed a significant positive relationship between $E_n$ and $E_d$, but the goodness of fit was better in 2020 ($R^2 = 0.80$) than in 2021 ($R^2 = 0.67$) after conducting boundary line analysis (Figure 4c,d). $E_n$ increased with increasing $E_d$, while it tended to be saturated when $E_d$ reached approximately 0.4–0.5 mm d$^{-1}$ in 2020 and 0.4 mm d$^{-1}$ in 2021.

### 3.3. Components of Nocturnal Transpiration and Xylem Refilling

$T_n$ was low and stable before mid-August, while it showed an obvious peak that appeared in mid-September; however, $R_e$ was relatively higher in May and remained stable between June and September in 2020 ($p < 0.000$, ANOVA) (Figure 5a and Table 1). In 2021, $T_n$ was low and stable between July and August, but showed peaks in early May and mid-September; however, $R_e$ in May and July was significantly lower than that in June and August ($p < 0.000$, ANOVA) and showed a peak in early September (Figure 5b and Table 1). The annual average $T_n$ values were 0.049 ± 0.073 and 0.052 ± 0.068 mm d$^{-1}$, while the $R_e$ values were 0.050 ± 0.038 and 0.043 ± 0.030 mm d$^{-1}$ in 2020 and 2021, respectively. The accumulated $T_n$ values over the growing season were 7.14 and 7.90 mm in 2020 and 2021, respectively, accounting for 49.76% and 54.44% of $E_n$. Meanwhile, the sums of $R_e$ were 7.22 and 6.61 mm, accounting for 50.24% and 45.56% of $E_n$ in 2020 and 2021, respectively.

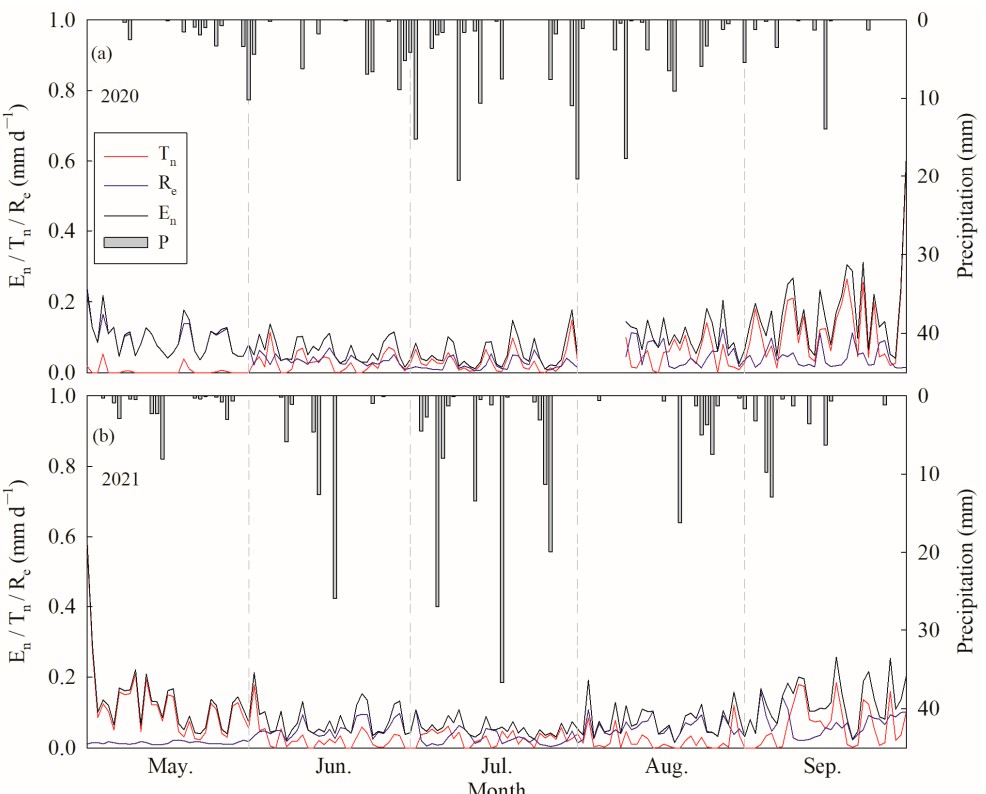

**Figure 5.** Daily variations of nocturnal water use ($E_n$), nighttime canopy transpiration ($T_n$) and stem refilling ($R_e$) and the distribution of precipitation during the growing season in (**a**) 2020 and (**b**) 2021.

**Table 1.** Seasonal variation in nocturnal water use ($E_n$), nocturnal transpiration ($T_n$), stem refilling ($R_e$) and contribution of $T_n$ and $R_e$ to $E_n$ ($T_n$:$E_n$ and $R_e$:$E_n$) during the growing season from May to September of 2020 and 2021 for *Caragana korshinskii* Kom.

|  | Month | $E_n$ (mm) | $T_n$ (mm) | $R_e$ (mm) | $T_n$:$E_n$ (%) | $R_e$:$E_n$ (%) |
|---|---|---|---|---|---|---|
|  | May | 3.05 | 0.14 | 2.91 | 4.72 | 95.28 |
|  | June | 1.95 | 0.89 | 1.05 | 45.86 | 54.14 |
| 2020 | July | 1.73 | 1.00 | 0.73 | 57.91 | 42.09 |
|  | August | 2.39 | 1.14 | 1.25 | 47.69 | 52.31 |
|  | September | 5.24 | 3.97 | 1.28 | 75.68 | 24.32 |
|  | May | 4.12 | 3.68 | 0.44 | 89.32 | 10.68 |
|  | June | 2.32 | 0.76 | 1.56 | 32.64 | 67.36 |
| 2021 | July | 1.74 | 0.89 | 0.85 | 50.96 | 49.04 |
|  | August | 2.34 | 0.61 | 1.73 | 26.07 | 73.93 |
|  | September | 3.99 | 1.96 | 2.02 | 49.25 | 50.75 |

### 3.4. Environmental and Stomatal Drivers of Nocturnal Water Use

The statistical analysis indicated that the significance effects of the environmental and stomatal variables on $E_n$ had a ranking of $G_{c-n} > u_{2-n} > T_{a-n} > PAR > RH_n$ in 2020 and $G_{c-n} > T_{a-n} > RH_n > PAR > u_{2-n}$ in 2021. Moreover, $T_n$ was positively correlated with $G_{c-n}$ and *PAR*, but negatively correlated to $T_{a-n}$, $u_{2-n}$ and $VPD_n$ in 2020; while it was positively correlated with $G_{c-n}$, and negatively correlated to $T_{a-n}$, $RH_n$ and $VPD_n$ in 2021, when controlling for $R_e$. However, $R_e$ was positively correlated with *PAR* and $VPD_n$ in both years, but negatively to $RH_n$ and $T_{a-n}$ in 2020 and to $RH_n$ and $u_{2-n}$ in 2021, when controlling for $T_n$ (Table 2).

**Table 2.** Correlation and partial correlation coefficients between nocturnal water use ($E_n$, $T_n$ and $R_e$) and environmental, stomatal variables for *C. korshinskii* in 2020 and 2021.

| | Controlling Variable | Biological Variables | | | | | |
|---|---|---|---|---|---|---|---|
| | | *PAR* | $T_{a-n}$ | $RH_n$ | $u_{2-n}$ | $VPD_n$ | $G_{c-n}$ |
| 2020 | $E_n$ | 0.455 ** | −0.496 ** | −0.183 ** | −0.532 ** | - | 0.748 ** |
| | Sig | 0.00 | 0.00 | 0.03 | 0.00 | - | 0.00 |
| | $T_n$ | 0.183 | −0.455 | - | −0.414 | −0.284 | 0.746 |
| | Sig | 0.02 | 0.00 | - | 0.00 | 0.00 | 0.00 |
| | $R_e$ | 0.458 | −0.332 | −0.543 | - | 0.366 | - |
| | Sig | 0.00 | 0.00 | 0.00 | - | 0.00 | - |
| 2021 | $E_n$ | 0.214 ** | −0.459 ** | −0.262 ** | −0.169 * | - | 0.658 ** |
| | Sig | 0.00 | 0.00 | 0.00 | 0.04 | - | 0.00 |
| | $T_n$ | - | −0.599 | −0.234 | - | −0.156 | 0.529 |
| | Sig | - | 0.00 | 0.00 | - | 0.05 | 0.00 |
| | $R_e$ | 0.285 | - | −0.313 | −0.375 | 0.440 | - |
| | Sig | 0.00 | - | 0.00 | 0.00 | 0.00 | - |

Note: *PAR* = photosynthetically active radiation; $T_{a-n}$ = nocturnal air temperature; $RH_n$ = nocturnal relative humidity; $u_{2-n}$ = nocturnal wind speed; $VPD_n$ = nocturnal vapor pressure deficits; $G_{c-n}$ = nocturnal canopy stomatal conductance; ** indicates $p < 0.01$, * indicates $p < 0.05$.

Both $E_n$ and $T_n$ decreased linearly with increasing $T_{a-n}$, which explained 87% and 99% of the variation in $E_n$, and 60% and 84% of the variation in $T_n$, during the measurement periods of 2020 and 2021, respectively. Meanwhile, $R_e$ was stable and decreased slightly with increasing $T_{a-n}$ in 2020, and increased linearly with increasing $T_{a-n}$ in 2021 (Figure 6a–c). Both $E_n$ and $T_n$ displayed an exponential decay response to $VPD_n$, but the goodness of fit was better ($R^2 = 0.72$, 0.93) in 2020 than in 2021 ($R^2 = 0.78$, 0.63). Meanwhile, $R_e$ exhibited a polynomial response to $VPD_n$ and tended to fall off when $VPD_n > 1.0$ kPa in 2020, but increased linearly with increasing $VPD_n$ (Figure 6d–f). $u_{2-n}$ explained 90%, 87% and 91% of the variation in $E_n$, $T_n$ and $R_e$ in 2020, and 47%, 32% and 93% in 2021, following exponential decay functions, respectively (Figure 6g–i). $E_n$ exhibited an exponential saturation response to $G_{c-n}$ both in 2020 and 2021, and $E_n$ tended to level off at 1.5 mm s$^{-1}$ in 2021. $T_n$ increased linearly with increasing $G_{c-n}$ in 2020, but had an exponential saturation response to $G_{c-n}$ in 2021. Meanwhile, $R_e$ remained stable with different $G_{c-n}$ values in 2020, but had an exponential decay response to $G_{c-n}$ in 2021 (Figure 6j–l).

Combined with the direct and indirect effects, $VPD_n$ positively affected $E_n$ (0.146) and $R_e$ (0.476), while it negatively affected $T_n$ (−0.135). $u_{2-n}$ decreased $E_n$ (−0.157) and $R_e$ (−0.273) during two growing seasons. Finally, the total effect of these five environmental and stomatal variables explained 50%, 36% and 32% of the variation in $E_n$, $T_n$ and $R_e$, respectively (Figure 7).

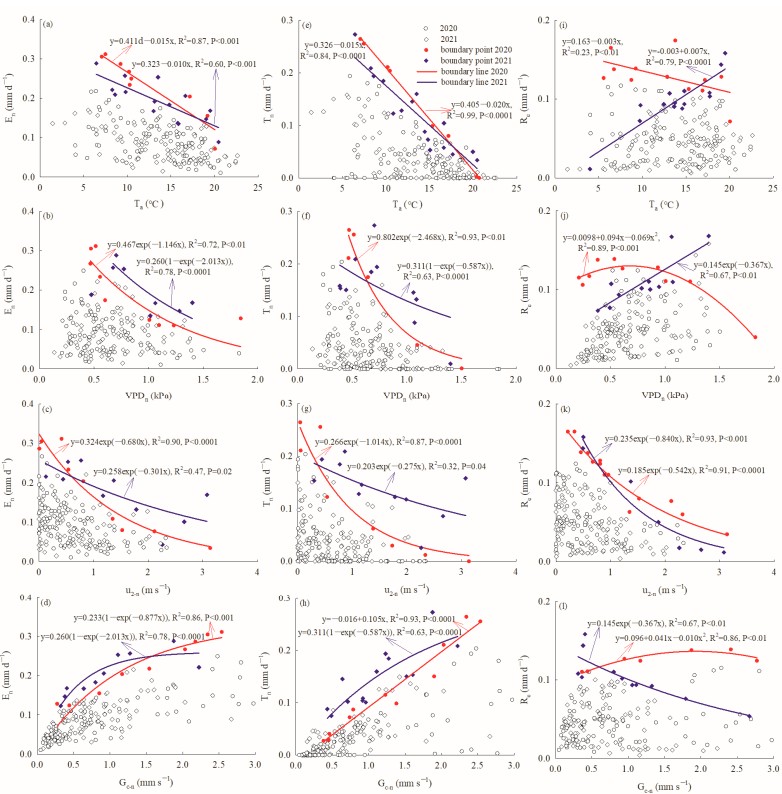

**Figure 6.** Nocturnal water use ($E_n$) relationship with (**a**) nocturnal air temperature, (**b**) nocturnal vapor pressure deficits, (**c**) nocturnal weed speed, (**d**) nocturnal canopy conductance; nocturnal transpiration ($T_n$) relationship with (**e**) nocturnal air temperature, (**f**) nocturnal vapor pressure deficit, (**g**) nocturnal weed speed, (**h**) nocturnal canopy conductance; stem refilling ($R_e$) relationship with (**i**) nocturnal air temperature, (**j**) nocturnal vapor pressure deficit, (**k**) nocturnal weed speed, (**l**) nocturnal canopy conductance.

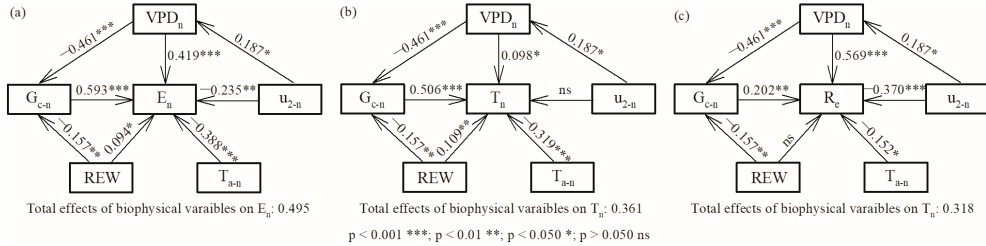

**Figure 7.** Direct and indirect effects of environmental and stomatal variables on (**a**) nocturnal water use, (**b**) nocturnal transpiration, and (**c**) stem refilling in 2020 and 2021.

## 4. Discussion

### 4.1. The Application of Thermal Dissipation Probes for C. korshinskii Plantation

Sap flow was usually measured using the stem heat balance method for shrubs with short height and small diameter at breast height (DBH) [28]. However, in arid and semiarid regions, the influence of high wind and strong solar radiation brought great challenges to the complete sap flow velocity data continuity and stability. The thermal dissipation probes (TDP10) method has recently been used to measure the sap flow velocity for plants with small DBH, such as bamboo species [40], *Populus adenopoda* [41], *Aegiceras corniculatum* [42] and so on. Compared with an induced hydraulic pressure and sap flow changing device and a whole-culm pot weighing method, the 10 mm long TDP was proven to be a valid means of measuring the sap flow of *Phyllostachys pubescens* [43]. Additionally, an environmental temperature reduction from 25 to 0 °C did not alter the values of $\Delta T_{max}$ between a heated probe and a reference probe when there was no sap flow, verifying that $\Delta T$ measured at



night can be used as a reference in daytime [44]. The daily average temperature in our study was $15.14 \pm 3.88$ °C and $15.02 \pm 4.28$ °C during the growing season in 2020 and 2021, respectively (Figure 3b). Therefore, TDP10 could be verified to be a valid means of accurately quantifying the water use of *C. korshinskii* with SDB less than 5 cm in this study.

*4.2. Nocturnal Water Use Behaviors of C. korshinskii*

Increasing evidence from leaf microstructure and gas exchange experiments has suggested that stomata are partially open throughout the night, which provides the necessary conditions for nocturnal water use [5,38]. Like other tree and shrub species, such as *P. euphratica*, *P. tabuliformis*, *A. truncatum*, *S. superba*, *Q. lancifolia*, *A. latifolia*, *A. jorullensis*, *S. psammophila*, we observed substantial *C. korshinskii* $E_n$ at night during the growing seasons in 2020 and 2021 (Figure 2). We found that the water stored in the stems, refiled mostly during the night from soil water uptake through the roots, is usually used for supply water consumption during the daytime [23,45]. However, the $E_n$ of *C. korshinskii* was much lower than that reported in plant species in the past. This may be related to the smaller daily transpiration of *C. korshinskii* plantations and the lower soil water content (Figure 3c) on the Bashang Plateau [31]. The $E_n{:}E$ values in this study were 14% and 13% in 2020 and 2021, respectively, which were similar to the values reported for *P. tabuliformis* and *A. truncatum* [22]. The sharp increase in $E_n{:}E$ at the beginning and the end of the growing season in both 2020 and 2021 was similar to that found in birch [45] and a *P. tomentosa plantation* [23]. Nocturnal water loss was more important under drought conditions than in other conditions [15]. The sharp increase in sap flow during the early growth stage had a function of refilling xylem conduits that accumulated embolisms during winter in the previous years. Additionally, air temperature during winter ranged from $-29$ °C to $-10$ °C in 2020 and $-30$ °C to $-5$ °C in 2021 (Figure 3b), the relatively higher $E_n{:}E$ was thus related to a minimized effect of freezing and drought on the hydraulic functioning of *C. korshinskii* at the beginning of the growing season [23]. Meanwhile, the sharp increase in soil water availability in September promoted an increase in $E_n$ and $E_n{:}E$, which was used to refill the embolized xylem conduits at the end of growing season. However, the seasonal trend of $E_n{:}E_d$ in our study was contrary to that reported by Zhao et al. [38]. Relatively smaller differences between $E_n$ and $E_d$ from June to August during the growing season resulted from smaller differences in diurnal air temperature and $T_{a-n}$ (Figure 3b).

*4.3. Nocturnal Transpiration and Stem Refilling Dynamics of C. korshinskii*

The partitioning of $E_n$ into nocturnal transpiration ($T_n$) and stem refilling ($R_e$) is challenging because of their obvious temporal overlap. The forecasted refilling method has been successfully and widely used in previous studies on rain-free days under low levels of $VPD_n$. However, Chen et al. [22] indicated that $E_n$ could also be influenced by nocturnal wind speeds, soil water content and their interaction. The accuracy of the forecasted refilling method can only be guaranteed under a low value of $VPD$. On every rain-free night in this study (Figures 2 and 3b), $VPD$ was usually lower than that reported in *A. truncatum* [5] and *P. euphratica* [46]; we thus ignored the uncertainty of the forecasting method in this study. Our results indicate that, for *C. korshinskii*, $T_n$ accounted for 49.76% and 54.44% of nocturnal water use, as the mean values of $R_e{:}E_n$ throughout the growing season were 50.24% and 45.56% in 2020 and 2021, respectively. The $R_e{:}E_n$ values were much higher than the 20%–25% observed for oak and pine growing in a humid montane region [27], and were similar to *S. superba* planted in a humid region in the dry season [20]; however, they were slightly lower than that found in *P. tomentosa* approximately 61% planted in a semiarid region [22], and much lower than that found in *P. euphratica* approximately 80% growing in an extremely arid environment [2] and *A. truncatum* (>85%) planted in an urban environment [5]. The higher value of $R_e{:}E_n$ in the water-limited environment indicated a greater reliance of plants on the water stored through stem refilling, which could help forests to maintain hydraulic support under higher transpiration demand [47,48]. This might be why canopy transpiration in *C. korshinskii* remained steady during the middle

and end of the growing season (Figure listed in Zhang et al. [31]). Moreover, compared with that occurring in 2021, a relatively higher $R_e$:$E_n$ (50.24%) value was observed in 2020, influenced by the relatively lower REW (Figures 3c and 5). Therefore, increasing $R_e$:$E_n$ could be an important drought adaptation strategy for forests to overcome seasonal water stress in the growing season.

*4.4. Environmental and Stomatal Effects on Nocturnal Water Use*

We found that nocturnal water use usually fluctuated with changes in the site's environmental and stomatal variables. We found negative relationships between $E_n$ and $T_{a-n}$, $VPD_n$ and $u_{2-n}$, which was consistent with previous studies [12,23]. $E_n$ was facilitated by $VPD_n$ when light was absent, especially for young species. However, associated with an increased risk of xylem cavitation and decreased hydraulic conductance within plant tissue, stomatal closure was also triggered by low SWC and atmospheric drought [49,50]. An obvious response of the $G_{c-n}$ threshold (1.5 mm s$^{-1}$) to $E_n$ was observed in both 2020 and 2021 (Figure 6j). Variations in canopy stomatal conductance were proven to be strongly correlated with variations in site SWC and VPD [51]; nocturnal water use was thus significantly affected by soil drought, especially under high-$VPD_n$ conditions. $E_n$ then declined in the face of high transpiration demands during the night (Figure 6d). Therefore, the negative relationship between $E_n$ and $VPD_n$ might be related to the physiological effects of $VPD_n$ on stomata [22]. $E_n$ decreased exponentially with increasing $u_{2-n}$ in this study, which was consistent with Gutiérrez et al. [52], but contrary to Zhao et al. [1]. The reasons for this could be as follows: (1) $u_{2-n}$ underwent an obvious change at night during the growing season at our study site (Figure 3a), and (2) $u_{2-n}$ mitigated $T_{a-n}$ inversion and finally suppressed $E_n$ by increasing $VPD_n$ [53]. Soil water availability is an important variable controlling nocturnal water use according to previous studies [9,12]. However, REW had a weak influence on $E_n$ in our study (Figure 7). Because soil water availability was very low surrounding the root system (Figure 3a), $E_n$ and $T_n$ were minimal when soil water content was low [21]. The response of $T_n$ to $VPD_n$ and $G_{c-n}$ was similar to that of $E_n$, but different to that of $R_e$ (Figure 6). A positive relationship between $R_e$ and the $VPD$ of previous day was reported for hybrid aspen coppice [6], indicating a high proportion of $SF_n$ in the refilling of dehydrated tissues. However, the driving force of stem refilling became weak with the decrease in water demand [54], especially under high levels of $VPD_n$. The difference in the environmental and stomatal mechanisms between $T_n$ and $R_e$ dynamics might be related to disequilibrium between leaf and soil water potentials and the whole tree's hydraulic conductance [6]. Therefore, analysis of plant hydraulic capacity in planted forests should be undertaken in the future.

## 5. Conclusions

In this study, nocturnal water use dynamics and their environmental and stomatal control mechanism were explored for a *C. korshinskii* plantation on the Bashang Plateau. Nocturnal water use, transpiration and stem refilling in *C. korshinskii* accumulated to 14.36, 7.14 and 7.22 mm in 2020 and 14.51, 7.90 and 6.61 mm in 2021, respectively. Nocturnal transpiration accounted for 49.76% and 54.44% of total nocturnal water use, while stem refilling accounted for 50.24% and 45.56%, which indicates that *C. korshinskii* was able to draw on water stored in the stem to overcome seasonal drought. The sharp increase in the ratio of nocturnal to diurnal water use that appeared at the beginning and the end of the growing season might be an ecological strategy for recovering the hydraulic conductivity of the xylem conduits. Nocturnal water use was negatively correlated with all meteorological variables, but increased with increasing nocturnal canopy conductance, which indicates that nocturnal water use was sensitive to stomatal regulation at night. Specifically, nocturnal water use was predominantly affected by nocturnal canopy conductance, nocturnal air temperature and nocturnal wind speed. In contrast, canopy conductance, nocturnal air temperature, and nocturnal vapor pressure deficits explained the highest variation in nocturnal transpiration, and nocturnal vapor pressure deficits and nocturnal wind speed

explained the highest variation in stem refilling. The total effects of the five environmental and stomatal variables explained 50%, 36% and 32% of the nocturnal water use, nocturnal transpiration and stem refilling variation, respectively. Our results provide a new understanding of water use strategies employed by plants in *C. korshinskii* plantations on the Bashang Plateau, and suggest that ecophysiological responses and adaptation to increasing drought severity and duration will occur under future climate changes.

**Author Contributions:** Conceptualization, W.L.; methodology, W.L. and Y.Z.; investigation, N.W. and Z.Q.; data curation, B.X. and C.L.; Supervision, J.C. and Y.Y.; Writing—original draft, W.L.; Writing—review and editing, W.L. and Y.Z. All authors have read and agreed to the published version of the manuscript.

**Funding:** This research was funded by the National Natural Science Foundation of China (No. 42001027, 42101019, 42371048), Science and Technology Project of Hebei Education Department (BJK2022022), the Key Research and Development Plan Project of Hebei Province (22324202D), Natural Science Foundation of Hebei Province (D2021403023), the Key Research and Development Plan Project of Ningxia Hui Autonomous Region (2021BEG02008), Funding for the Science and Technology Innovation Team Project of Hebei GEO University (KJCXTD-2021-10), Innovation and Entrepreneurship Training Program for College Students (202210077015, S202210077021), and Hebei GEO University Student Science and Technology Fund (KAG202303).

**Data Availability Statement:** The data presented in this paper are available on request from the corresponding author.

**Conflicts of Interest:** The authors declare no conflict of interest.

## Appendix A

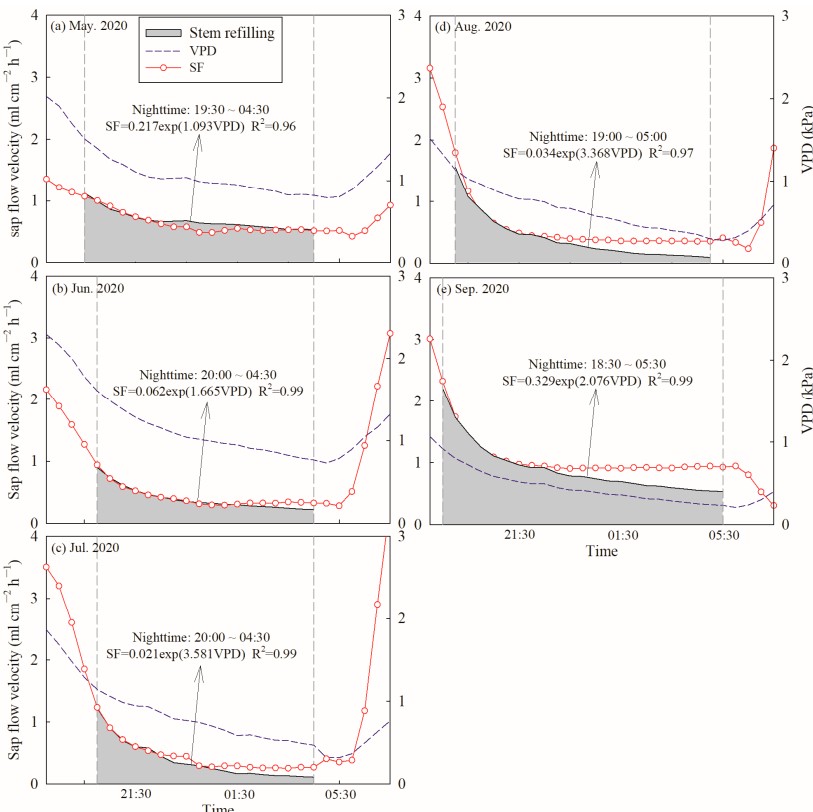

**Figure A1.** Examples of the forecasted refilling method used to determine the percentage of nocturnal sap flow velocity attributable to nocturnal transpiration and stem refilling in *C. korshinskii*. Nighttime was defined between dashed gray vertical lines (PAR < 5 μmol m$^{-2}$ s$^{-1}$). Stem refilling is the shade proportion of nocturnal sap flow velocity, (**a**) May in 2020, (**b**) June in 2020, (**c**) July in 2020, (**d**) August in 2020 and (**e**) September in 2020.

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
