# Peer review of "Nocturnal Water Use Partitioning and Its Environmental and Stomatal Control Mechanism in Caragana korshinskii Kom in a Semi-Arid Region of Northern China"

_forests, doi:10.3390/f14112154_

Round 1

Reviewer 1 Report

Comments and Suggestions for Authors

1 The average radius of the sample trees was estimated to be 1.3 cm (estimated based on the average As of each stem), while the length of TDP sensor is 2 cm in length. the 2cm TDP sensor covered one radius and another inner part (0.7 cm in length) of cross-sectional area, so there will induce error when the sap flow density is not even across the sapwood considering radial variation of sap density is common in various tree species. it may be better to use short TDP sensor for the sap flux measurement of shrubs.

2 In line 160-161, the authors cited the empirical equation used by Granier, while the equation expressed in line 161 is not same as the original formula of Granier.

3 In line 198, the estimation of Gc should base on nocturnal transpiration (Tn) instead of E (the total of Tn and Re).

4 In line 211-212, the linear and nonlinear regression analysis were expressed, while in the Fig.4 and Fig.6 the authors showed boundary analysis, please show the results of regression analysis and present why use boundary analysis if necessary.

Comments on the Quality of English Language

The language of this manuscript needs to be improved for clearer understand.

Reviewer 2 Report

Comments and Suggestions for Authors

Zhang et al. investigated xylem sap flow in a Caragana korshinskii plantation in a semi-arid region of northern China. The use of sap flow sensors to not only assess water transport but also quantify water use is urgently needed to assess the water and carbon balance of forest stands, especially under a changing climate. While the approach is promising, I have a few questions/comments that need to be addressed before publication:

-The plant size was very small (stem diameter about 2.6 cm), while the installed sensors were also 20 mm long. Can the authors estimate or calculate how e.g. air temperature might have affected the measurements?

- Most important point: The calculations of xylem sap flow need to be clarified. What ΔTmax was used in the calculations? As there is a shift in ΔTmax during the growing season, it needs to be recalculated from time to time. This is not clear in the manuscript at the moment, but it would strongly influence the calculations for nocturnal water flux.

- How did you account for repeated measurements when comparing daily/monthly data? (L206-207) This is not explained yet.

- The figures are very blurred and unreadable.

- The results in sections 3.3 and 3.4 can only be assessed once the sap flow calculations have been clarified. The same applies to the discussion.

For detailed comments see the attached pdf.

Comments on the Quality of English Language

Often sentences are overly long and should be shortened. 

Reviewer 3 Report

Comments and Suggestions for Authors

This study monitored the nighttime water use of the Caragana korshinskii shrub in a semi-arid region of northern China during 2020-2021, and partitioned nighttime sap flow into nighttime transpiration (water loss via stomata) and stem refilling. Then further assessed the biophysical controls of nighttime sap flow. The study seems well-designed and the topic is interesting, but I cannot fully evaluate the presented results due to the poor quality of figures, which must be improved to increase readability.

My main concern goes to the methodology. This study used the 2-cm Granier type TDP sensors on a relatively small shrub species with an average stem diameter of 2.38 cm, the authors did not describe how they installed the sensors exactly, but I doubt drilling and inserting of 2-cm sensors on those small branches/stems of only ~2.5 cm diameter could be very challenging and most importantly may likely cause wounding effects, which introduce great measurement uncertainties. Besides, the sensors were installed only 15 cm above the ground, which means the ground natural temperature gradient may greatly influence the TDP measurements. I am afraid these two caveats may greatly compromise the results presented here.

Comments on the Quality of English Language

English writing is generally fine, but there are apparent grammatical and spelling issues that need to be addressed. 

Round 2

Reviewer 3 Report

Comments and Suggestions for Authors

I appreciate the efforts the authors put into their revisions. While my main concern remains. Installation of the 2-cm TDP probes on such small-sized stems (~3 cm) would highly likely cause a notable wounding effect which could introduce great measurement errors, and I am not fully convinced that the effects of the ground temperature regime would be negligible by only applying aluminum foil wrapping + sparse understory grass cover.  Even in dense forests with a close canopy and dense understory vegetation, sap flow practitioners conventionally install sensors at a diameter at breast height (~1.3 m) to avoid ground temperature influences. The ground temperature influences in your study site could be potentially greater than a forest site due to more open canopy, sparse vegetation, and thus great solar radiation. You may consider setting up some sort of ground solar radiation reflection to avoid this uncertainty in your future studies.

Author Response

We have studied the Academic Editor’ comments carefully and tried our best to revise our manuscript according to the comments: (1) the section of 2.2.1 Materials, We have added detail information of the nine sampled stems of C. korshinskii in our study, such as Line 132-136,  “Based on the distribution of stem base diameter (SBD), nine sample stems (SDB = 2.68, 2.96, 3.38, 3.62, 3.66, 3.76, 4.14, 4.24, and 4.38 cm) of C. korshinskii plantation…”; (2) We have revised the methods to reflect the updated information in the part of 2.2.3 Sap flow measurements. Thermal dissipation probes used in this study was TDP10, and the detail information of insulation protection was added, such as Line 160-170, “Nine pairs of Granier-type probes (TDP10, Dynamax Inc., TX, USA) consisting of a cop-per-constantan thermojunction were inserted at a depth of 10 mm into the xylem sapwood of stem of nine shrubs at height of 40 cm above the ground on the northern side…”; (3) We have added some information of ∆Tmax correction in Line 177-185, and we applied a double regression method to recalculate the real “∆Tmax” in this study, such as “∆Tmax (℃) is the maximum temperature difference between sensors, which was determined as the maximum value of daily ∆Tmax over a 9-day period to avoid underestimation of the night-time sap flow…”; (4) We have added some discussion appropriate probe types for smaller plants in the section of 4.1, “The application of thermal dissipation probes for C. korshinskii plantation” (Line 362-378). All of the revised information was marked with red colour in the revised manuscript. 
